# Stem Cell Origin of Cancer: Biological Principles and Clinical Strategies for Chemoprevention and Maintenance Therapy in Cancer Care

**DOI:** 10.3390/cancers17162621

**Published:** 2025-08-11

**Authors:** Yusra Medik, Sehrish Sardar, Jaskirat S. Sethi, Marcelo P. Bigarella, Sunny R. K. Singh, Shi-Ming Tu

**Affiliations:** 1Division of Hematology and Oncology, University of Arkansas for Medical Sciences, Little Rock, AR 72205, USA; 2Department of Urology, University of Arkansas for Medical Sciences, Little Rock, AR 72205, USA

**Keywords:** chemoprevention, cancer stem cells, tumor microenvironment, GLP-1R, metformin, maintenance therapy

## Abstract

When we prevent cancer, we no longer need to cure it. We propose that formulation of a pertinent theory about the origin and nature of cancer is imperative for the discovery of innovative and impactful as well as safe and affordable chemoprevention measures including anti-inflammatory and/or anti-insulinemic agents, which modulate cancer stem cell (CSC) functions and attenuate signal pathways/networks and the microenvironment of both progenitor CSCs and progeny differentiated cancer cells.

## 1. Introduction


*An Ounce of Prevention Is Worth a Pound of Cure.*
-Benjamin Franklin

There is evidence that cancer stem cells (CSCs) play an active role in carcinogenesis [1,2]. How CSCs mimic and mirror normal stem cells and may be related to if not derived from them has important biological and clinical implications. Interestingly, all cancer hallmarks, including heterogeneity, metastasis, dormancy, drug resistance, and immunity, are deeply immersed in stem-ness functions and activities. However, there is still a glaring gap and a missing link in our basic understanding between a stem cell origin of cancer and the prevention of its occurrence or maintenance of its remission in cancer care.

A basic premise in this Perspective is that if cancer were a stem cell disease with stem-ness properties, then judicious management of its putative CSCs and their respective onco-niches would prevent the initiation and preempt the promotion of carcinogenesis. Therefore, a unifying theme in the prevention of cancer is to keep our normal stem cells pristine and cancer stem cells restrained. It is imperative that we treat an individual patient and the entire multicellular cancer with its various compartments, myriad components, and the microenvironments.

In this article, we revisit a brief history of and highlight recent advances in cancer chemoprevention. We focus on chemoprevention measures that modulate CSC expressions and attenuate signal pathways/networks of both progenitor CSCs and progeny differentiated cancer cells. We wonder about embryonic and oncogenic biomarkers, such as transforming growth factor-beta (TGF-β) and insulin-like growth factor (IGF)-1, and ponder about glucagon-like peptide-1 receptor (GLP1-R) agonists in the prevention of cancer. We propose to harness the power of chemoprevention ingrained in our diet and invigorated by our microbiome.

## 2. Brief History

A comprehensive history of cancer prevention is both enlightening and cautionary [3]. In 1727, Le Clerc recommended removing tumors (including polyps) before they become cancerous [4]. In 1775, Pott recognized that protective clothing might shield chimney sweepers from soot exposure and prevent scrotal cancer [5].

In 1976, Sporn introduced the term chemoprevention for cancer prevention [6]. However, the Selenium and Vitamin E Cancer Prevention Trial (SELECT) suggested that micronutrient supplementation might not prevent certain cancers [7]. Although the Prostate Cancer Prevention trial (PCPT) showed that finasteride reduced prostate cancer risk, it did not affect high-grade, clinically significant disease [8].

In the 1980s, smoking cessation and screening/resection revolutionized our culture and practice of cancer prevention [3]. In 1993, sulindac, a nonsteroidal anti-inflammatory drug, was demonstrated to decrease the incidence of colorectal adenomas [9]. Importantly, high-dose aspirin selectively prevented cyclooxygenase-2 expressing colorectal cancer [10]. Similarly, hepatitis B vaccine lowers risk for liver cancer [11], while human papillomavirus vaccine reduces that for cervical cancer [12].

History has taught us that there may be a universal and unified theory of cancer’s origin and nature that addresses all aspects of CSCs, including heterogeneity, metastasis, drug resistance, cancer immunity, cancer dormancy, and empowers us to design and implement improved cancer prevention strategies and modalities—to prevent cancer’s initiation and promotion at its very seed and roots whether it is cellular versus genetic, and infectious or metabolic [1,2].

One way to prevent cancer is to avoid conditions or behaviors that instigate or aggravate it. History has proven to us the efficacy of refraining from cigarette smoking, alcohol dependency, and other noxious exposures that damage stem cells and overwhelm their ability to repair and heal, to harmonize and equilibrate. It has also shown us the supremacy of the scientific method to elucidate exceptional clinical observations and validate pertinent scientific hypotheses in our conduct of cancer research and provision of cancer care [13].

## 3. Stem-Cell Theory

Although Virchow had already postulated a stem cell theory of cancer in 1874 [14], the hypothesis of a stem cell origin of cancer is still evolving and in need of reconciling with other prevalent popular cancer theories, such as a genetic origin of cancers [15].

A genetic origin of cancer in essence is reductionistic. It epitomizes acquisition and accumulation of genetic mutations as the cause of cancers. Hence, to prevent cancer, we avoid acquisition and accumulation of these mutations. Better still, we fix them or target them in our therapeutics. It spurs targeted therapy and spawns precision medicine.

A stem cell origin of cancer in contrast is integrative. Although genetic content is vital, epigenetic context is pivotal. It accounts for intercellular interactions and microenvironmental interplays. Hence, to prevent cancer, one needs to tame the cells and coax the niches. It prioritizes multimodal therapy and emphasizes integrated medicine.

Let us take the example of preventing prostate cancer. A central idea derived from a stem cell origin of cancer is that prostate cancer comprises differentiated cancer cells that depend on testosterone, utilize androgen receptor (AR) and AR pathways, and CSCs that do not. This explains the findings of increased incidence of low-risk prostate cancer (odds ratio 1.61) but not high-risk prostate cancer (odds ratio 0.44) within one year of testosterone replacement therapy [16]. Similarly, both finasteride in the PCPT trial [17] and dutasteride [18] in the REDUCE trial reduced risk of Gleason ≤ 6 but not high-grade prostate cancers.

According to a stem cell origin of prostate cancer, preventing potentially lethal prostate cancer is just as important if not more important than incidental prostate cancer [19]. Without a proper cancer hypothesis, we may end up treating the right targets in the wrong cells and often enough, obtaining equivocal results and attaining marginal clinical benefits.

## 4. Stem-Cell Therapy

A stem cell theory of cancer predicates that there should be stem cell biomarkers with prognostic values and stem cell targets with therapeutic implications in cancer care.

It is uncanny that carcinogenesis shares the same stem-ness genes and employs the same epithelial-to-mesemchymal transition pathways (such as wnt/beta catenin, Snai1/Zeb1/Hh) as embryogenesis does [20,21,22]. Importantly, targeting stem-like pathways has therapeutic implications, because stem-ness may be the true driver, if not engine, of the malignant process. Indeed, anti-stem-like activity elicits anti-cancer effects (e.g., anti-HER2, anti-Nectin4, anti-Trop2, anti-PDL1) for a variety of cancers [23], because stem-ness feature is more than likely to be a universal (i.e., agnostic) property of cancer than not.

Ironically, research on induced pluripotent stem cells (iPSC) [24,25] for the development of stem-cell therapy may have provided us some inadvertent insights about a stem cell origin of cancer.

For instance, Abad et al. [26] managed to induce pluripotency in a variety of cell types within living mice using a recipe of four “stem-ness” factors (namely, OCT4, SOX2, KlLF4, and c-MYC). They showed that expression of certain stem-ness genes is produced iPSC but also teratomas. Importantly, this seminal experiment provided indisputable proof that stem-ness properties (i.e., inherent in iPSC) was the cause, if not the source, of cancer, i.e., teratoma formation.

Similarly, Ohnishi et al. [27] demonstrated that transient expression of reprogramming factors in vivo resulted in tumor development in various tissues. Because iPSC generation did not require changes in genomic sequence, their data challenged the traditional belief that cancer arose primarily through accumulation of genetic mutations. Again, their results aligned more with a stem cell than a genetic theory of cancer in which cellular state (i.e., as manifested in iPSC) trumps genetic drivers.

In addition, Nori et al. [28] evaluated long-term safety of iPSC-based cell therapy in a spinal cord injury model. They found that engrafted differentiated iPSC led to functional recovery and stimulated synapse formation at 47 days after transplantation. However, long-term observation (for up to 103 days) revealed deteriorated motor function due to tumor formation derived from distinct human-iPSC clones in this mouse model.

## 5. Vitamin D and Stem Cells

In 1919, Mellanby speculated that rickets developed in the absence of some dietary factors and that a “4th vitamin” could prevent this deficiency disease [29]. Nowadays, a deficiency in this “4th vitamin”, namely vitamin D, still occurs in some medical conditions (e.g., cystic fibrosis, celiac disease, obesity) and with certain medications (e.g., laxatives, steroids, phenytoin). Because vitamin D deficiency is associated with a variety of health problems, including cancer, cardiovascular disease, diabetes, autoimmune diseases, and depression [30], it is plausible that vitamin D also affects stem cell health in one way or another.

Indeed, the precursor of vitamin D3 cholecalciferol stimulates cellular proliferation, expression of pluripotency markers (NANOG, SOX2, and OCT4), and the osteogenic differentiation potential of bone marrow-derived mesenchymal stem cells (MSC), while it reduces senescence [31].

Hence, Perez et al. [32] showed that imbalances in various hormones affect the health of stem cells in their respective endocrine organs. Intriguingly, vitamin D (2000 IU/d for 5.3 years) prevented death by almost 20% from advanced cancer (defined as metastatic or lethal), even though it did not prevent overall cancer in a large, randomized “VITAL” trial (the Vitamin D and Omega-3 Trial) [33]. This is reminiscent of the PCPT trial (as mentioned above in Section 2) [8], in which control of progenitor cancer stem cells vs. progeny differentiated cancer cells may have differential chemoprevention effects on the development of metastatic or lethal vs. overall prostate cancer, respectively.

Furthermore, the clinical benefit of vitamin D for the prevention of advanced cancer (38% decrease) was limited to people who were normal weight than among those who were overweight or obese. Perhaps obesity causes not only insulin but also vitamin D resistance, because the body no longer responds to either insulin or vitamin D, respectively. Perhaps obesity affects stem-ness and the stem-like microenvironment, as well as progenitor stem cells vs. progeny differentiated cells, in which insulin and vitamin D no longer function effectively and appropriately.

## 6. Niche Matters

Recently, Mascharak et al. [34] showed that patterns of scar tissue in a tumor are among the most predictive and prognostic (second only to stage at diagnosis) for pancreatic ductal carcinoma. The architecture and organization of cells in the scar tissue could foretell a median difference in survival of almost 2 years. This is remarkable considering that the 5-year survival rate of patients with ductal pancreatic cancer is only 20–25%.

Specifically, they investigated the extracellular matrix pattern of 437 patients and integrated it with spatial mapping of immune and stromal cells. They found that tumors with more disordered, sheet-like matrix structures and enriched pro-inflammatory cell interactions had significantly worse outcomes. Notably, certain fibroblast and B cell spatial niches correlated with distinct desmoplastic patterns and survival.

What remains unclear is whether the desmoplasia in a tumor scar that displays an interplay among various cellular and stromal constituents alludes to distinct tumor subtypes with different cellular vs. genetic origins and phenotypes. In other words, does the presence of pro-inflammatory fibroblasts, activated B cells, and densely packed fibers versus cytotoxic T cells and thin fibers within tumors indicate separate stem-ness origins and their inseparable onco-niches?

After all, the immune system provides us striking examples of the niche in life and death, in health and disease. When it concerns immune activation and immune suppression, pro-inflammation and anti-inflammation, involvement of MSC or macrophages, interactions among progenitor stem cells or progeny differentiated cells and their respective niches, it is not a matter of good or bad.

When we are under attack by viruses, pro-inflammatory forces promote immune-stimulatory responses. Activated immune cells such as T cells and natural killer (NK) cells produce interferon-gamma (IFN)-γ, which increases major histocompatibility complex (MHC) expression and enables differentiation of stem cells. The MSC and macrophages are in a pro-inflammatory mode. There are more antigen-presenting cells (APC) cells in the fray. The goal is to eliminate the viruses, as well as contain the damages caused by the viruses.

Once the viral attack has subsided, anti-inflammatory forces modulate immune-suppressive responses. The same agents assume an opposite role and mode to calm the immune system and clamp inflammation. Time to withdraw the troops. Time to dampen the fire. IFN-γ now decreases MHC expression and attenuates differentiation of stem cells. The MSC and macrophages are in a recovery and healing phase. The APC cells can now leave. The goal is to ensure that the troops do not overreact, and the fire does not overtake them.

Importantly, this dualism in nature has therapeutic implications, because a particular niche is not necessarily good or bad depending on the appropriate cellular context. Chemoprevention measures may be effective and beneficial for one tumor subtype or phenotype under certain conditions but counterproductive and even detrimental for a different subtype or phenotype in a different clinical setting.

## 7. Riddle of TGF-β

An enduring riddle in cancer biology is the pervasiveness of its “yin-yang” dichotomy. How can it be that TGF-β is both a tumor suppressor and promoter? This dualistic characteristic of TGF-β is emblematic in cancer research and problematic for cancer care. To solve this puzzle is to uncover the origin of cancer.

TGF-β is a tumor suppressor because it mediates anti-proliferative and pro-apoptotic effects. It is a vital stromal factor promoting cellular differentiation and inhibiting cellular mobilization in a variety of tissues including the bone and lungs.

TGF-β is also a tumor promoter because it induces tumor motility, invasion, metastasis, and epithelial-to-mesenchymal transition. The “switch” of TGF-β from suppressor to promoter seems to occur in one way or another during malignant transformation in many cancers.

However, a narrative of cancer depends on our perspective of cancer. According to the genetic theory of cancer, TGF-β makes or breaks cancer. When we focus on TGF-β in a reductionist view of cancer, it is the main if not the sole actor that determines cancer. However, according to a stem cell theory of cancer, TGF-β is but one of many team players in a game of cancer. In an integrated view of cancer, its exact role is dependent on the proper cellular context, i.e., timing and time, placing and place of its activity and action.

Hence, TGF-β is a different actor and player in the fetus versus an adult, during embryogenesis versus carcinogenesis. What it does in the fetus may be perfectly benign during embryogenesis, but the same activity and action in an adult can be patently malignant during carcinogenesis [23].

Again, this is another instance of oncology recapitulating ontogeny. When embryogenic factors reemerge and resurge, carcinogenic processes become manifest and rampant. The paradox of TGF-β as both a tumor suppressor and promoter may be one key to unlock the stem cell theory of cancer. If cancer has a stem cell origin and is a stem cell disease, it does not need to reprogram or reinvent itself, it does not need to hijack what it already owns or retrieve what it does not need.

Therefore, the clinical implications of a correct cancer theory cannot be more obvious in cancer research and for cancer care. We predict that a stem cell origin (unified theory) of cancer will largely determine whether targeting TGF-β in drug vs. therapy development for chemoprevention specifically and for anti-cancer therapy in general will be a success or failure [13].

## 8. Dualism of IGF-1

IGFR-1/IGF-1 is another example of the two faces of Janus when it resembles a tumor suppressor in one cellular context but behaves like a tumor promoter in another. Perhaps a proper cancer theory will help us solve this problem of dualism for the purpose of chemoprevention in cancer care.

Interestingly, people with Laron syndrome and IGF-1 deficiency have a low incidence of cancer [35]. Although many of them are obese and live in deprived environments, they also have a decreased incidence of cardiovascular disease due to their better insulin sensitivity and lower blood pressure.

It is well known that IGF-1R is a classical survival signaling factor, and IGF-1R-expressing stem cells display robust multipotent properties [36]. Furthermore, lower IGF-1 level favors increased expression of IGF-1R. However, since IGF-1R regulates stem cell multipotency and reprogramming in both normal and cancer stem cells [37], separating the two may be requisite but problematic in cancer care.

Intriguingly, inhibition of the IGF-1/IGF-1R axis decreases the number of regulatory T cells (Tregs), M2 macrophages, myeloid-derived suppressor cells, and enhances the recruitment and activity of M1 macrophages, dendritic cells, cytotoxic T cells, and NK cells. Since IGF-1R plays an essential role in protecting stem cells and preventing autoimmunity, anti-IGF-1R may disrupt immune tolerance and unwittingly elicit autoimmune complications [38,39].

Hopefully, a stem cell origin and unified theory of cancer will enable us to design appropriate chemoprevention strategies that favorably modulate rather than ignorantly muddle the IGF-1/IGF-1R axis to enhance both efficacy and safety for our patients [40].

## 9. Dilemma with Testosterone

Currently, we have a dilemma with testosterone in the care of patients with prostate cancer. Since 1941, androgen deprivation therapy (ADT) has been a mainstay of advanced prostate cancer therapy. However, there is a plethora of evidence (since 1996) suggesting that androgens can also suppress prostate cancer growth [41]. Hopefully, a proper cancer theory about the effects of testosterone on prostate CSCs vs. differentiated cancer cells will enable us to reconcile these contradictory observations.

After all, prostate CSCs do not express AR and do not need testosterone, whereas differentiated prostate cancer cells do [42]. This fact is consistent with finding showing that testosterone replacement therapy is associated with an increase in favorable-risk prostate cancer but a lower risk of aggressive prostate cancer [16]. Understandably, a pertinent theory about the origin and nature of cancer influences how we design chemoprevention and maintenance therapy in the care of patients with prostate cancer.

Recently, Chen et al. [43] demonstrated that canonical androgen response element motifs are tumor suppressive regulatory elements in the prostate. Safi et al. [44] showed that androgen receptor monomers and dimers regulate opposing biological processes in prostate cancer cells. Alarmingly, ADT may induce neuroendocrine differentiation and promote the emergence of a more aggressive, life-threatening castration-resistance prostate cancer with neuroendocrine phenotypes [45,46].

Importantly, clinical trials have verified and validated the merits of androgen replacement vs. deprivation therapy in appropriate clinical settings [47,48,49]. Therefore, a paradox of testosterone in prostate cancer development could be a paradigm in prostate cancer treatment, when it concerns the optimal design of an intermittent ADT regimen—instead of patients receiving testosterone, they produce it naturally when they are off ADT.

For example, one could design an intermittent ADT clinical trial that targets minimal residual disease comprising prostate CSCs after maximal response to ADT of 6–9 months and then prolong the period off ADT for as long as possible by allowing testosterone to recover to normal levels while keeping PSA to as low as possible using a maintenance regimen that suppresses prostate CSCs (see Section 8) [50].

Therefore, a correct cancer theory about the origin and nature of cancer impacts how we prevent cancers and maintain remissions. According to a stem cell (unified) theory of cancer (Figure 1), ADT eliminates the bulk of AR-positive, PSA-producing, differentiated prostate cancer cells, whereas maintenance therapy controls the few remaining AR-negative, non-PSA producing prostate CSCs and their microenvironment, which would not and should not be adversely affected by testosterone replacement or testosterone recovery off ADT on an intermittent schedule.

## 10. Ideal Chemoprevention Agents?

Recently, O’Connor et al. [52] reported an association between metformin use and reduced cancer incidence in a variety of malignancies. Among its myriad mechanisms of actions, metformin decreases IGF-1, increases AMPK, mitigates inflammation, modulates the intestinal microbiota, empowers DNA repair, enhances autophagy, inhibits cellular senescence, and inactivates reactive oxygen species [53,54].

If there is a simple reason for metformin’s versatile anti-cancer and various chemoprevention capabilities, perhaps it is that they all converge on stem cell health [55]. After all, metformin restores normal stem cell function and minimizes normal stem cell exhaustion. Conversely, metformin may prevent the conversion of normal stem cells to CSCs and keeps both more healthy—stable, inactive, and harmless—than not.

Not surprisingly, GLP-1R agonists (GLP-1Ra) also reduced the risk of 10 obesity-associated cancers, namely esophageal, colorectal, endometrial, gall bladder, renal, hepatic, ovarian, pancreatic, meningioma, and multiple myeloma (but not gastric, postmenopausal breast, and thyroid) in patients with type 2 diabetes and treated with GLP-1Ra versus insulin [56]. Curiously, there was no decrease in cancer risk when comparing GLP-1Ra with metformin in this retrospective, observational study.

Similarly, even though patients lost significantly more weight after bariatric surgery compared to GLP-1Ra at 1–2 years, both reduced the risk of obesity-related cancers (by 19%) of 13 cancers, including colorectal, pancreatic, breast, ovarian, and liver cancers compared with no intervention [57]. Importantly, GLP-1Ra intervention was associated with lower all-cause mortality (HR 0.5) compared with bariatric surgery (HR 0.86).

Interestingly, GLP-1Ra was associated with reduced colorectal cancer (CRC) risk in drug-naive patients with type 2 diabetes with and without obesity/overweight, with more profound effects in patients with obesity/overweight, suggesting a potential protective effect against CRC was partially mediated by weight loss as well as by other mechanisms not related to weight loss [58].

Finally, if GLP-1Ra was able to delay cancer recurrence, then it would have the capacity to control minimal residual disease that contained refractory malignant clones (likely comprise CSCs) and could be utilized as a maintenance treatment to prolong remission [59]. Therefore, when it concerns stem cell health vs. harm, both chemoprevention strategies and maintenance treatments target CSCs to achieve their respective favorable, if not superior, clinical outcomes.

## 11. All Roads Lead to Stem-Ness

Perhaps it is not a coincidence that both metformin and GLP-1Ra enable people to lose weight and prevent cancer: metformin increases secretion of GLP-1 [60,61], while GLP-1Ra redoubles its effects. We propose that all roads to cancer prevention including those that relate to obesity or euglycemia, and those that operate through TGF-β, IGF-1, or GLP-1—lead to stem-ness or stem-like properties in normal and/or cancer stem cells.

It makes sense that GLP-1/R exerts anti-inflammatory and immune suppressive effects through Treg and peripheral T cells and anti-inflammatory cytokines, such as IL-4, IL-10, and TGF-β to attenuate auto-inflammatory activities and prevent pathological immune responses, as well as inhibiting the release of pro-inflammatory cytokines, such as IL-2, IFN-γ, and tumor necrosis factor-α [62].

What one may not yet appreciate is that GLP-1Ra and metformin may also reduce mortality and cardiovascular events and ameliorate other comorbid conditions by way of reducing inflammation and protecting healthy stem cells and mediating cancer chemoprevention effects by counteracting against cancer stem cells [63,64,65].

For example, GLP-1/Notch signaling is involved in stem cell maintenance [66,67]. GLP-1Ra provides protective effects on the cardiovascular system by improving CD34+ hematopoietic stem progenitor cell functions [68]. GLP-1 also modulates pluripotency/differentiation of MSC [69] and adipose-derived stem cells [70].

It is of interest that natural products may modulate CSC expressions and attenuating signal pathways/networks of progenitor CSCs vs. progeny differentiated cancer cells by way of their anti-inflammatory and/or anti-insulinemic activities.

For example, diverse dietary bioactive components, e.g., curcumin, lycopene, and others exert their chemoprevention and anti-tumor effects in part through modulation of miRNA expression, such as miR-34 and let-7 that result in an indolent stem-ness/stem-like if not an innocuous CSC phenotype [71,72,73].

Indeed, many natural histone deacetylase inhibitors [74] found in food, e.g., diallyl disulfide (in garlic and onions), sulforaphane-rich extracts (in broccoli and other cruciferous, cabbage-family vegetables), polyphenols (in turmeric, berries, soy, green tea), and butyrate modulate and control CSCs and keep them innocuous, if not inert, and can be taken for a prolonged period of time in a safe manner.

We may not yet have a bird’s eye view of all the roads that lead to stem-ness in cancer research and in cancer care. However, many of us have already arrived at a crossroad of cancer prevention in which a unified cancer theory about a stem-cell vs. genetic origin of cancer may help us navigate so that cancer research will be more efficient and informative and cancer therapeutics will be more effective and less risky in the foreseeable future (Table 1).

## 12. Conclusions

In this Perspective, we discuss a unified theory and stem cell origin of cancer, as envisioned by Virchow, and examine its clinical implications in cancer prevention and cancer care. We ruminate about the role of the tumor microenvironment, vitamin D, and GLP-1R in chemoprevention and maintenance therapy.

According to a stem cell theory of cancer, chemoprevention is effective when we do not perturb or provoke either CSCs or non-CSCs. Therefore, ensuring that CSCs remain dormant and indolent while assuring that non-CSCs stay pristine and healthy make sense.

Surely, Hygieia, the Greek goddess of health (Figure 2) and daughter of Asclepius, the god of medicine, would be pleased with and proud of Benjamin Franklin for cancer prevention. After all, he died at age 84 years and outlived all of his 16 siblings and 3 children during a time when the average life span was only 35–38 years [75], and he did not die from cancer.

## Figures and Tables

**Figure 1 cancers-17-02621-f001:**
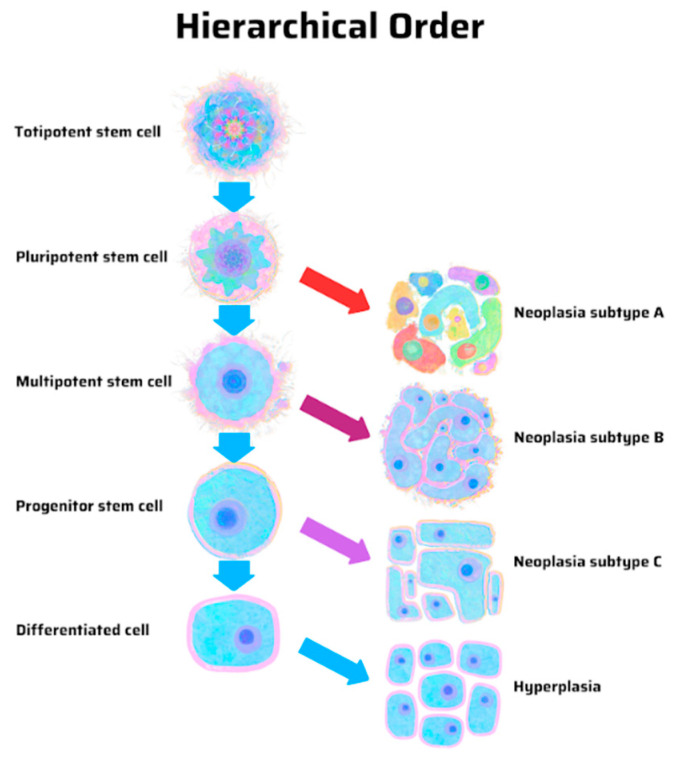
Origin of cancer in a hierarchical order, according to the stem cell theory of cancer. Unique cancer subtypes with specific phenotypes arise from distinct progenitor cancer-initiating cells with tapering stem-ness properties. Neoplasia subtype/phenotype A is more likely to be heterogeneous or mixed and contain both cancer stem cells and differentiated cancer cells than neoplasia subtype/phenotype C does. Reprinted with permission from Tu SM (2021) [51]. Copyright 2021 MDPI.

**Figure 2 cancers-17-02621-f002:**
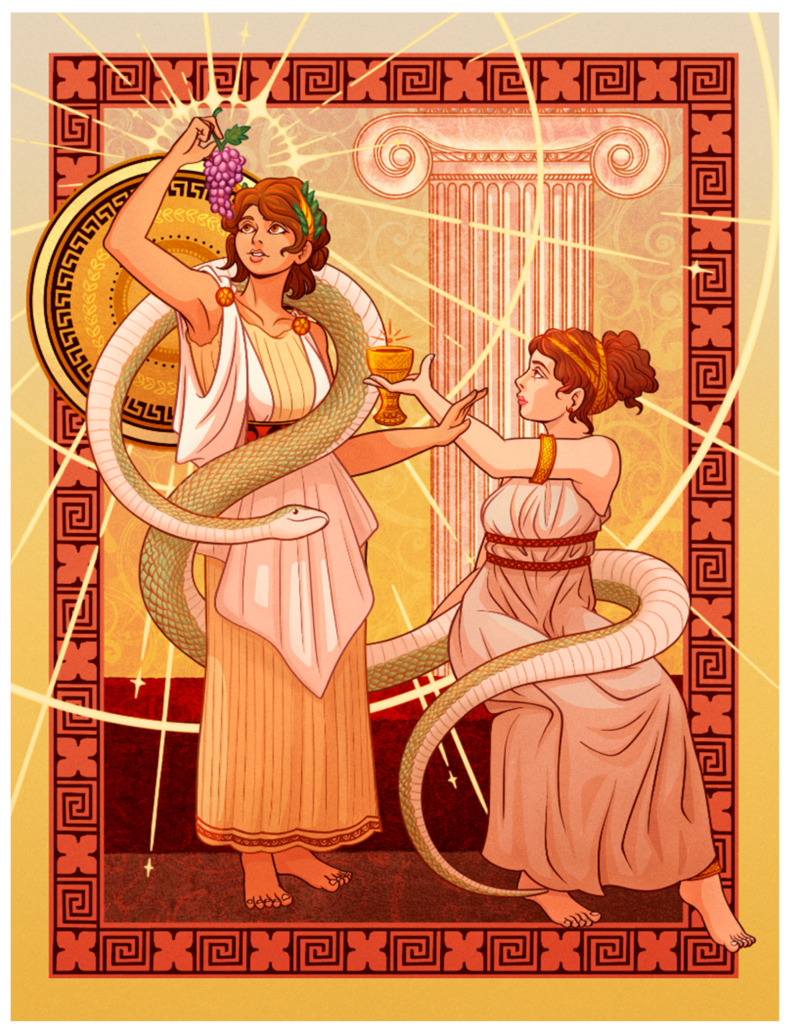
An artist’s rendition of Hygieia, the Greek goddess of prevention and her sister, Panacea, goddess of cure (sitting and holding an elixir). Behind her is a shield, symbol of protection. Their bodies (“stems”) are encircled by a snake (representing renewal, healing, and resurrection, as snakes shed their skin like the body can shed disease and return to health). Created by and with permission from Benjamin Tu (www.bentubox.com) for this article on 20 May 2025.

**Table 1 cancers-17-02621-t001:** Summary of selected anti-cancer agents and targets for the purposes of chemoprevention and maintenance therapy in cancer care.

Anti-Cancer Agent	Cancer Initiation/Promotion	Cancer Prevention (Stem-Ness or -Like Targets and Mechanisms)	Targeted Therapies, Pertinent Clinical Trials
AR inhibitor		Non-stemness (or -like) cells	Finasteride, PCPT trial [8]
Vitamin D		Stemness (or -like) cells Reduces senescence [23]	Vitamin D and Omega-3 trial (VITAL) [25]
TGF-beta	Cancer promotion Induces motility, invasion, EMT	Tumor suppression Anti-proliferation; mediates apoptosis	Bintrafusp alfa [13]
IGF-1/R	Cancer stem cells Proinflammatory, immune-stimulatory Onco-niche	Stemness (or -like) cells Anti-inflammatory, immune-inhibitory [29] “Embryonic” niche [30]	Cixutumumab [33]
Metformin		Decreases IGF-1, mitigates inflammation, enhances autophagy, inhibits cellular senescence, inactivates ROS [46,47], induces GLP-1 [53,54], maintains stem cell integrity [48]	Cancer prevention and maintenance therapy [48]
GLP-1R		Stemness (or -like) cells [59,60,61,62,63] Anti-inflammatory, immune-inhibitory [55]	Semaglutide, tirzepatide [49,50,51,52]

AR, androgen receptor; TGF, transforming growth factor; IGF-1/R, insulin-like growth factor-1/receptor; ROS, reactive oxygen species; GLP-1R, glucagon-like peptide-1 receptor; EMT, epithelial-to-mesenchymal transition.

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
