# Peer review of "Stem Cell Origin of Cancer: Biological Principles and Clinical Strategies for Chemoprevention and Maintenance Therapy in Cancer Care"

_cancers, 2025, doi:10.3390/cancers17162621_

Round 1
Reviewer 1 Report
Comments and Suggestions for Authors
In this commentary the authors discuss the stem cell origin of cancer theory as a basis for cancer chemoprevention. This article follows on the heels of a similar commentary published last year in the same journal by the same authors that addresses immunotherapy in the context of the stem cell origin of cancer theory. In the present manuscript the authors outline their premise that the cancer stem cell model predicts that factors/agents that can suppress the stemness of cancer stem cells while preserving the properties of normal tissue stem cells could be used in preventing cancer. After discussing a history of cancer prevention dating back to the 1700s the authors provide a very brief introduction to CSCs, and then discuss vitamin D, environmental niche, obesity, and the vexing properties of TGFb, IGF1 and testosterone in cancer. The authors conclude by returning to the idea that cancer stem cells make for a logical target for chemoprevention and maintenance therapy.
Overall the manuscript comes across as a stream of consciousness rather than an unbiased and rigorous argument for targeting the cancer stem cell population. Moreover, the authors appear to focus on suppressing the properties of CSCs as a preventative, and do not acknowledge strategies that might selectively eradicate CSCs without harming stem or differentiated normal tissue cells. Overall, the manuscript would benefit from more in-depth analysis and commentary in each of its sections. For example:
- The history section is hardly comprehensive and comes across as a discussion of random achievements while largely disregarding current research in the prevention field. This section would benefit from greater depth.
- Likewise, the background section on CSCs lacks significant depth.
- The illustration of Hygieia and Panacea add little to the readers’ understanding of the arguments. A figure illustrating the authors’ arguments and conclusions would be more suitable.
Author Response
We thank the reviewers for your invaluable comments and suggestions. We have provided our responses below in bold with changes highlighted in red underneath their comments. We have incorporated all changes with the changes highlighted in red in the revised manuscript.
In this commentary the authors discuss the stem cell origin of cancer theory as a basis for cancer chemoprevention. This article follows on the heels of a similar commentary published last year in the same journal by the same authors that addresses immunotherapy in the context of the stem cell origin of cancer theory. In the present manuscript the authors outline their premise that the cancer stem cell model predicts that factors/agents that can suppress the stemness of cancer stem cells while preserving the properties of normal tissue stem cells could be used in preventing cancer. After discussing a history of cancer prevention dating back to the 1700s the authors provide a very brief introduction to CSCs, and then discuss vitamin D, environmental niche, obesity, and the vexing properties of TGFb, IGF1 and testosterone in cancer. The authors conclude by returning to the idea that cancer stem cells make for a logical target for chemoprevention and maintenance therapy.
Overall the manuscript comes across as a stream of consciousness rather than an unbiased and rigorous argument for targeting the cancer stem cell population. Moreover, the authors appear to focus on suppressing the properties of CSCs as a preventative, and do not acknowledge strategies that might selectively eradicate CSCs without harming stem or differentiated normal tissue cells. Overall, the manuscript would benefit from more in-depth analysis and commentary in each of its sections.
Thank you for your succinct summary of this Perspective article, in which we aimed to provide some breadth and depth to the stem cell theory of cancer as a basis for cancer chemoprevention.
As a perspective rather than a review article that aligns with the main premises of a unified theory of cancer as specified in the Special Issue, we offered a viewpoint with examples and references so that the ideas are sufficiently focused for those who prefer a big picture and also for those who look for some details. For instance, once we acknowledge that prostate cancer comprises both non-CSC and CSC (section 9), treating the former with ADT is effective but not curative, while treating the latter with metformin and GLP-1Ra (section 10) could be preventive and prolong remission after ADT.
For example:
- The history section is hardly comprehensive and comes across as a discussion of random achievements while largely disregarding current research in the prevention field. This section would benefit from greater depth.
The purpose of the section on a “Brief History” is to summarize seminal observations/research (which were not random achievements in our opinion) that have paved the way for and as an introduction to current research (e.g., O’Connor, JNCI 2024 [52]; Wang, JAMA Netw Open 2024, [56]; and, Lin, JCO 2024 [57]) in cancer prevention, as dicussed in later sections of this Perspective. We have adhered to the ideals of brevity and unity as expected in a Perspective article and as recommended by other reviewers.
- Likewise, the background section on CSCs lacks significant depth.
Thank you for your invaluable comments and suggestions! We have dedicated a more indepth section to Stem Cell Theory and separated it from the section on Stem Cell Therapy.
- Stem cell Theory
Although Virchow had already postulated a stem cell theory of cancer in 1874 (3), the hypothesis of a stem cell origin of cancer is still in dispute because stem cells are difficult to study and because there is a need to reconcile it with other prevalent popular cancer theories, such as a genetic origin of cancers (4).
A genetic origin of cancer in essence is reductionistic. It epitomizes acquisition and accumulation of genetic mutations as the cause of cancers. Hence, to prevent cancer, we avoid acquisition and accumulation of these mutations. Better still, we fix them or target them in our therapeutics. It spurs targeted therapy and spawns precision medicine.
A stem cell origin of cancer in contrast is integrative. Although genetic content is vital, epigenetic context is pivotal. It accounts for intercellular interactions and microenvironmental interplays. Hence, to prevent cancer, one needs to tame the cells and coax the niches. It prioritizes multimodal therapy and emphasizes integrated medicine.
Let us take the example of preventing prostate cancer. A central idea derived from a stem cell origin of cancer is that prostate cancer comprises differentiated cancer cells that depend on testosterone, utilize androgen receptor (AR) and AR pathways, and CSC that do not. This explains the findings of increased incidence of low-risk prostate cancer (odds ratio 1.61) but not high-risk prostate cancer (odds ratio 0.44) within one year of testosterone replacement therapy (16). Similarly, both finasteride in the PCPT trial (17) and dutasteride (18) in the REDUCE trial reduced risk of Gleason <6 but not high-grade prostate cancers.
According to a stem cell origin of prostate cancer, preventing potentially lethal prostate cancer is just as important if not more important than incidental prostate cancer (19). Without a proper cancer hypothesis, we may end up treating the right targets in the wrong cells and often enough, obtaining equivocal results and attaining marginal clinical benefits.”
In addition, we believe that the references in the background section on CSC have provided sufficient breadth and depth without losing focus on this Perspective article (it is not a Review article after all). Specifically, we condensed the whole idea of a stem cell origin of cancer from two books (refs 1 and 2) that discussed about all aspects of CSC including heterogeneity, metastasis, drug resistance, cancer immunity, cancer dormancy (hence, a unified theory of cancer), and which together provided about 800 references on this topic. In particular, ref 1 has a chapter on Cancer Stem Cells (pages 67-81 with 23 references).
- The illustration of Hygieia and Panacea add little to the readers’ understanding of the arguments. A figure illustrating the authors’ arguments and conclusions would be more suitable.
We have added Figure 1 to illustrate the main ideas and reinforce the arguments and conclusions in this Perspective.
(see revised manuscript)
Figure 1. Origin of cancer in a hierarchical order, according to the stem cell theory of cancer. Unique cancer subtypes with specific phenotypes arise from distinct progenitor cancer-initiating cells with tapering stem-ness properties. Neoplasia subtype/phenotype A is more likely to be heterogeneous or mixed and contains both cancer stem cells and differentiated cancer cells than neoplasia subtype/phenotype C does. Adapted from Cancers with permission [51].
The illustration of Hygieia and Panacea (sister goddesses) reminds us about a special relationship between prevention and cure that may be missed when we take care of cancer patients and perform cancer research.
Reviewer 2 Report
Comments and Suggestions for Authors
Medik et al have written a perspective on stem cells and their targeting for chemoprevention and maintenance for cancer treatment. The perspective is well-written and covers an interesting topic. I have a few suggestions
- Adding a cancer stem cell theory diagram will increase the presentation.
- When we talk about chemoprevention in cancer by targeting SCs, natural products play a significant role. Some references can be added, highlighting their importance.
- A brief note on SC signaling pathways would be beneficial.
Author Response
We thank the reviewers for your invaluable comments and suggestions. We have provided our responses below in bold with changes highlighted in red underneath their comments. We have incorporated all changes with the changes highlighted in red in the revised manuscript.
Medik et al have written a perspective on stem cells and their targeting for chemoprevention and maintenance for cancer treatment. The perspective is well-written and covers an interesting topic.
Thank you for your kind comments!
I have a few suggestions
- Adding a cancer stem cell theory diagram will increase the presentation.
Done.
Figure 1. Origin of cancer in a hierarchical order, according to the stem cell theory of cancer. Unique cancer subtypes with specific phenotypes arise from distinct progenitor cancer-initiating cells with tapering stem-ness properties. Neoplasia subtype/phenotype A is more likely to be heterogeneous or mixed and contains both cancer stem cells and differentiated cancer cells than neoplasia subtype/phenotype C does. Adapted from Cancers with permission [51].
(See revised manuscript)
- When we talk about chemoprevention in cancer by targeting SCs, natural products play a significant role. Some references can be added, highlighting their importance.
Thank you for your excellent suggestion and insights! Added a paragraph (before last) to Section 11, “All Roads Lead to Stemness”:
It is of interest that natural products may modulate CSC expressions and attenuate signal pathways/networks of progenitor CSC vs progeny differentiated cancer cells by way of their anti-inflammatory and/or anti-insulinemic activities.
For example, diverse dietary bioactive components, e.g., curcumin, lycopene, and others exert their chemoprevention and anti-tumor effects in part through modulation of miRNA expression, such as miR-34 and let-7 that result in an indolent stem-ness/stem-like if not an innocuous CSC phenotype [71-73].
Indeed, many natural histone deacetylase inhibitors [74] found in food, e.g., diallyl disulfide (in garlic and onions), sulforaphane-rich extracts (in broccoli and other cruciferous, cabbage-family vegetables), polyphenols (in turmeric, berries, soy, green tea), and butyrate temper CSC and keep them inactive, if not dormant, and can be taken for a prolonged period of time in a safe manner.
- A brief note on SC signaling pathways would be beneficial.
Added brief note to Section 4, Stem Cell Therapy, paragraphs 1 and 2.
A stem cell theory of cancer predicates that there should be stem cell biomarkers with prognostic values and stem cell targets with therapeutic implications in cancer care.
It is uncanny that carcinogenesis shares the same stem-ness genes (such as N-cadherin, SNAIL, ZEB) and employs the same epithelial-to-mesenchymal transition pathways (such as WNT/beta catenin, Hedgehog) as embryogenesis does [20-22]. Importantly, targeting stem-like pathways has therapeutic implications, because stem-ness may be the true driver, if not engine, of the malignant process. Indeed, anti-stem-like activity elicits anti-cancer effects (e.g., anti-HER2, anti-Nectin4, anti-Trop2, anti-PDL1) for a variety of cancers [23], because stem-ness feature is more than likely to be a universal (i.e., agnostic) property of cancer than not.
Reviewer 3 Report
Comments and Suggestions for Authors
This perspective manuscript is related to stem cell-associated tumor initiation and progression. The manuscript is well-organized and -written; however I have some concerns before its publication.
1. Why did you choose to write a perspective article rather than a review article? Please clarify how your manuscript fits the scope and purpose of a perspective piece, especially in terms of presenting a novel, hypothesis-driven viewpoint.
2. I suggest that the authors include a discussion of the clonal and hierarchical models of tumor heterogeneity.
3. There are many known anti-cancer agents. Why did you choose to include only those presented in Table 1? Please clarify the selection criteria.
Author Response
We thank the reviewers for your invaluable comments and suggestions. We have provided our responses below in bold with changes highlighted in red underneath their comments. We have incorporated all changes with the changes highlighted in red in the revised manuscript.
This perspective manuscript is related to stem cell-associated tumor initiation and progression. The manuscript is well-organized and -written; however I have some concerns before its publication.
- Why did you choose to write a perspective article rather than a review article? Please clarify how your manuscript fits the scope and purpose of a perspective piece, especially in terms of presenting a novel, hypothesis-driven viewpoint.
Because the scope and purpose of our manuscript is more commentary than comprehensive and is mainly concerned with the presentation of a novel, hypothesis-driven viewpoint, we chose to write it as a perspective rather than a review article.
- I suggest that the authors include a discussion of the clonal and hierarchical models of tumor heterogeneity.
Thank you for your excellent suggestion!
We discussed stem cell origin of cancer and heterogeneity and illustrated a clonal and hierarchical model of tumor heterogeneity in Figure 1:
Introduction, section 1, lines 33-35: Interestingly, all cancer hallmarks including heterogeneity, metastasis, dormancy, drug resistance, immunity, are deeply immersed in stem-ness functions and activities.
Brief History, section 2, lines 69-71: there may be a universal and unified theory of cancer’s origin and nature that addresses all aspects of CSC, including heterogeneity, metastasis, drug resistance, cancer immunity, cancer dormancy
Stem Cell Theory, section 3, lines 94-97: A central idea derived from a stem cell origin of cancer is that prostate cancer comprises differentiated cancer cells that depend on testosterone, utilize androgen receptor (AR) and AR pathways, and CSC that do not.
Stem Cell Theory, section 3, lines 102-103: According to a stem cell origin of prostate cancer, preventing potentially lethal prostate cancer (neoplasm subtype or phenotype A) is just as important if not more important than incidental prostate cancer (neoplasm subtype or phenotype C).
Figure 1. Origin of cancer in a hierarchical order, according to the stem cell theory of cancer. Unique cancer subtypes with specific phenotypes arise from distinct progenitor cancer-initiating cells with tapering stem-ness properties. Neoplasia subtype/phenotype A is more likely to be heterogeneous or mixed and contains both cancer stem cells and differentiated cancer cells than neoplasia subtype/phenotype C does. Adapted from Cancers with permission [51].
(See revised manuscript)
- There are many known anti-cancer agents. Why did you choose to include only those presented in Table 1? Please clarify the selection criteria.
Table 1 summarizes the results of those pertinent measures of cancer prevention and remission maintenance based on control of both CSC and differentiated cancer cells, according to a stem cell origin and unified theory of cancer, as discussed in this Perspective article.
In our original paper, we did talk about how many known anti-cancer agents (including natural products) played a significant role in cancer chemoprevention by targeting SCs.
However, the reviewers felt the paper was too long and overly diversified, and recommended that we separated it into two papers to ensure brevity and unity, as expected in a Perspective article.
We have reorganized the second manuscript, “Stem Cell Origin of Cancer: Clinical Implications for Chemoprevention and Maintenance Therapy in Cancer Care” that included the following sections: “Nutritional Intervention”, “Chemoprevention: CSC Targets”, “More Anti-CSC Agents” (manuscript in preparation) to supplement and complement with the current one.
Importantly, we have added a brief note to Section 4, “Stem Cell Therapy”, paragraphs 1 and 2 for the sake of being comprehensive and inclusive of a discussion about SC signal pathways to account for efficacy of and potential clinical benefits from other known anti-cancer agents when it concerns chemoprevention and maintenance therapy in cancer care:
A stem cell theory of cancer predicates that there should be stem cell biomarkers with prognostic values and stem cell targets with therapeutic implications in cancer care.
It is uncanny that carcinogenesis shares the same stem-ness genes (such as N-cadherin, SNAIL, ZEB) and employs the same EMT pathways (such as WNT/beta catenin, Hedgehog) as embryogenesis does [20-22]. Importantly, targeting stem-like pathways has therapeutic implications, because stem-ness may be the true driver, if not engine, of the malignant process. Indeed, anti-stem-like activity elicits anti-cancer effects (e.g., HER2, Nectin4, Trop2, PDL1) for a variety of cancers, because stem-ness features may be a universal (i.e., agnostic) property of cancer [23].
Likewise, we have added a paragraph (before last) to Section 11, “All Roads Lead to Stemness” for the purposes of being inclusive if not comprehensive of other known anti-cancer agents when it concerns chemoprevention and maintenance therapy in cancer care:
It is of interest that natural products may modulate CSC expressions and attenuating signal pathways/networks of progenitor CSC vs progeny differentiated cancer cells by way of their anti-inflammatory and/or anti-insulinemic activities.
For example, diverse dietary bioactive components, e.g., curcumin, lycopene, and others exert their chemoprevention and anti-tumor effects in part through modulation of miRNA expression, such as miR-34 and let-7 that result in an indolent stem-ness/stem-like if not an innocuous CSC phenotype [71-72].
Indeed, many natural histone deacetylase inhibitors [73] found in food, e.g., diallyl disulfide (in garlic and onions), sulforaphane-rich extracts (in broccoli and other cruciferous, cabbage-family vegetables), polyphenols (in turmeric, berries, soy, green tea), and butyrate modulate and control CSC and keep them innocuous, if not inert, and can be taken for a prolonged period of time in a safe manner.
Round 2
Reviewer 1 Report
Comments and Suggestions for Authors
The revised manuscript is suitable for publication.
Author Response
The revised manuscript is suitable for publication.
We thank the reviewer for your insightful suggestions for this manuscript!
Reviewer 3 Report
Comments and Suggestions for Authors
Dear Authors!
Thank you for taking into consideration my comments.
I think your manuscript is suitable for publication.
Author Response
Thank you for taking into consideration my comments. I think your manuscript is suitable for publication.
We appreciate your insightful comments that have improved this manuscript.